# Finding Optimal Arms in Non-stochastic Combinatorial Bandits with Semi-bandit Feedback and Finite Budget

**Jasmin Brandt**[a], **Viktor Bengs**[b], **Björn Haddenhorst**[a], **Eyke Hüllermeier**[b,c]

[a]Department of Computer Science, Paderborn University, Germany
[b]Institute of Informatics, University of Munich (LMU), Germany
[c]Munich Center for Machine Learning, Germany

jasmin.brandt@upb.de, viktor.bengs@lmu.de, bjoernha@mail.upb.de, eyke@lmu.de

## Abstract

We consider the combinatorial bandits problem with semi-bandit feedback under finite sampling budget constraints, in which the learner can carry out its action only for a limited number of times specified by an overall budget. The action is to choose a set of arms, whereupon feedback for each arm in the chosen set is received. Unlike existing works, we study this problem in a non-stochastic setting with subset-dependent feedback, i.e., the semi-bandit feedback received could be generated by an oblivious adversary and also might depend on the chosen set of arms. In addition, we consider a general feedback scenario covering both the numerical-based as well as preference-based case and introduce a sound theoretical framework for this setting guaranteeing sensible notions of optimal arms, which a learner seeks to find. We suggest a generic algorithm suitable to cover the full spectrum of conceivable arm elimination strategies from aggressive to conservative. Theoretical questions about the sufficient and necessary budget of the algorithm to find the best arm are answered and complemented by deriving lower bounds for any learning algorithm for this problem scenario.

## 1 Introduction

The multi-armed bandits (MAB) problem is an intensively studied problem class in the realm of machine learning, in which a learner is facing a sequential decision making problem under uncertainty [32]. A decision (action) corresponds to making a choice between a finite set of specific choice alternatives (objects, items, etc.), also called *arms* in reference to the metaphor of gambling machines in casinos. After each decision to choose a particular arm, the learner receives some form of *feedback* – typically a numerical reward – determined by a *feedback mechanism* of the chosen arm. The learner is not aware of the arms' feedback mechanisms and consequently tries to learn these in the course of time by performing actions according to its learning strategy. The concrete design of its learning strategy depends essentially on two main components of the learning setting: the assumptions on the feedback mechanisms and the learning task.

Traditionally, and even up to now the most prevalent assumption is that the feedback received by choosing one arm is generated by means of a probability distribution of the chosen arm [41, 31]. In this way, any useful learning strategy revolves around learning specific probabilistic features of the arms' distributions such as the means. These features, in turn, quite naturally provide a way to define a notion of *(sub-)optimality of an arm* as well as a *best arm*. A relaxation of this *stochastic setting* is the *non-stochastic setting*, in which no assumption is made in the form of probabilistic laws of the feedback mechanisms. Instead, either no assumptions are made on the feedback mechanisms, so that

36th Conference on Neural Information Processing Systems (NeurIPS 2022).

these can also be generated by an adversary [7], or that the sequence of feedback observations (or a transformation thereof) of an arm converges asymptotically to a fixed point [25]. In the latter case, the notion of an arm's (sub-)optimality is again straightforward, given that the limit points can be ordered, while in the former usually the best action in hindsight plays the role of the best arm.

Regarding the learning task, the most prominent one is that of *regret minimization*, where in each learning round the learner suffers regret unless the optimal decision is made (determined by the feedback mechanisms). The main challenge for the learner is to manage the trade-off between exploration and exploitation, i.e., constantly balancing (i) the degree of new information acquisition about some arms' feedback mechanism in order to appropriately expand the current knowledge base (exploration), and (ii) the degree of choosing arms considered to be optimal given the current knowledge base in order to keep the overall regret low (exploitation). In many practical applications, however, the learning task is of a quite different kind, as the focus is rather on finding the (approximately) correct answer to a problem specific question, e.g., which arm is the (approximately) optimal one, within a reasonable time (number of learning rounds). This *pure exploration* learning task can be considered in two variants, namely the fixed confidence and the fixed budget setting. In the former the learner tries to find the answer within as few as possible learning rounds, while guaranteeing a given pre-determined confidence for the correctness of its returned answer. In the latter, it is the other way around, as the learner is provided with a limit on the possible number of learning rounds (budget) and the confidence for the returned answer should be as high as possible. In both variants the main challenge for designing a suitable learner is to specify a clever exploration strategy for finding the correct answer.

In order to model more complex learning settings in practice, the basic setup of MAB problems has been generalized in various ways, such as incorporating additional side information [3, 6, 17, 2] or infinite number of arms [11, 38], just to name a few. Of special practical interest is the generalization of the basic setup, where the learner is allowed to choose specific sets of arms as its action. Consider as an example the online algorithm selection problem with parallel runs, where for sequentially arriving problem instances one selects a subset of available algorithms (solvers) to be run in parallel in order to solve the current problem instance.

If the feedback received is of a numerical nature, this variant has manifested itself under the term *combinatorial bandits* [14] while for feedback of a qualitative nature this variant can be referred to as *preference-based bandits* as put forward by [8]. Combinatorial bandits are further distinguished with respect to the type of feedback between semi-bandit feedback, where feedback of each single arm in the selected subset is observed, and full bandit feedback, where only some aggregated value of the individual numerical feedback of the arms in the selected subset is observed. Although both combinatorial and preference-based bandits consider a similar action set and for both the learner needs to deal with the possibly exponential size of the action set, the process of learning is quite different due to the nature of the observed feedback. The main reason for this is that in preference-based feedback the mutual correlations that may exist between the arms in the chosen subset play a major role in both the assumption about the feedback mechanisms and the learning task from the outset. In contrast to this, the standard setting in combinatorial bandits with semi-bandit feedback is that the individual reward generation mechanisms are independent of the chosen subsets. However, this modeling assumption is questionable in a couple of practical applications, especially when humans provide the feedback. For example, in opinion polls or rating systems where humans rate a subset of objects (political parties/candidates, products, etc.), it is well known that the ratings of the objects may be affected by context effects, i.e., preferences in favor of an object may depend on what other objects are available. In the fields of economics and psychology, context effects are among others divided into compromise [47], attention [22] and similarity [48] effects.

In this paper, we take a step towards unifying these two variants for the best arm identification (BAI) problem in a pure exploration learning setting with fixed budget and non-stochastic feedback mechanisms. The main motivation for this unification is to derive a general purpose learner, which can tackle the BAI problem in both feedback variants. In this way, for example, one can transform a learning problem with numerical signals into a preference-based learning problem and thus conveniently apply such a general purpose learner. Recent works have demonstrated in two different learning scenarios with numerical feedback that such a transformation has great potential [37, 27].

Needless to say, the main challenge is to unify both feedback variants through suitable abstractions allowing them to be treated as instantiations of the same problem class. This bridge is built by dropping

the common independence assumption (of the chosen arm set) for the numerical combinatorial bandits and abstracting the nature of the observations. Additionally, we simply assume that the learner is provided with an appropriate statistic customized to the explicit nature of the feedback. By appropriate choice of the statistic one obtains the respective setting, e.g., the empirical mean for the case of numerical feedback and relative frequencies for the case of preference feedback.

**Our contribution.** Under mild assumptions on the asymptotic behavior of these statistics, we derive a proper definition of a best arm a learner seeks to find (Section 2) as well as lower bounds on the necessary budget for this task (Section 3). To the best of our knowledge, such lower bounds are novel for non-stochastic settings and the derivation is rather non-standard due to the combinatorial setup of the problem. We suggest a general algorithmic framework suitable to cover the full spectrum of conceivable arm elimination strategies from aggressive to conservative, which we analyze theoretically regarding the algorithms' sufficient and necessary budget to find the best arm (Section 4). As a consequence, we obtain to the best of our knowledge the first algorithm(s) for non-stochastic preference-based bandits as well as for combinatorial bandits under semi-bandit feedback, in which the individual (numerical) feedback received for an arm depends on the chosen subset due to possibly existing mutual correlations between the arms in the chosen subset. The mild assumptions on the asymptotics of the statistics allow to transfer our theoretical results to the stochastic counterparts of the semi-bandit combinatorial and preference-based bandits (Section 5). We demonstrate the usefulness of the generality of our setting in an experimental study for an algorithm selection problem with parallel runs (Section 6), where once again the transformation of numerical feedback to preference feedback plays a key role. Additional experiments are given in the supplementary material, where also all proofs of the theoretical results are collected.

**Related Work.** A large body of literature considers the combinatorial bandit problem under preference-based feedback, see [8] for an overview. Although *dueling bandits* [52] has established as an overall agreed term for the scenario with actions of size two, the terminology for action sets of larger sizes is still discordant, e.g., multi-dueling [10], battling [42], choice [4] or preselection bandits [9], mainly due to subtle nuances of the motivating practical applications. While pure exploration settings with a stochastic preference-based feedback haven been considered by a series of works [36, 39, 15, 40, 43, 21], a pure exploration setting under a non-stochastic feedback mechanism as in our case has yet to be studied.

Pure exploration has been intensively studied in the basic multi-armed bandits (MAB) setting with stochastic feedback mechanisms as well, see Section 33.5 in [32] for a detailed overview. The non-stochastic variant of the fixed budget MAB setting is considered in [25], which is the backbone for the well-known Hyperband algorithm [34] and additionally inspired in some part the assumptions we make for our work. Initiated by the work of [12] to design learners for regret minimization frameworks which can perform well in both stochastic and non-stochastic settings, the fixed budget framework has been the subject of research by [1] and [45].

Combinatorial bandits with numerical feedback have been introduced by [14] and [16] in a regret minimization framework. The fixed confidence setting for stochastic combinatorial bandits with semi-bandit feedback is studied in [26], and full bandit feedback in [18, 30]. Finally, the best-of-$k$ bandits game introduced in [46], which in some way unifies the combinatorial bandits with binary set-dependent feedback and preference-based bandits in one joint framework similarly as we do in this work. However, they consider a fixed confidence setting with stochastic feedback mechanisms and do not provide a learner for the dependent arm case, although they derive lower bounds on the worst case sample complexity for this case. The only work assuming a set-dependent feedback mechanism in combinatorial bandits with semi-bandit feedback is in [50], where, however, the regret minimization task is studied under stochastic feedback mechanisms. In summary, there seems to be no existing work which considers a pure exploration setting for combinatorial bandits with non-stochastic or even stochastic semi-bandit feedback, where the (mean) rewards of the arms in the chosen subset of arms depends on the subset. Accordingly, our results provide new theoretical contributions to this field.

## 2 Problem Formulation

In our setup, we assume a set $\mathcal{A}$ of $n$ arms, which we simply identify by their indices, i.e., $\mathcal{A} = [n] = \{1, \ldots, n\}$. For some fixed $k < n$ we denote the set of all possible subsets of arms with size of at least 2 and at most $k$ by $\mathcal{Q}_{\leq k} = \{Q \subseteq \mathcal{A} \mid 2 \leq |Q| \leq k\}$. Further, we assume that for any

$Q \in \mathcal{Q}_{\leq k}$ we can query some feedback, in the form of a feedback vector $\mathbf{o}_Q = (o_{i|Q})_{i \in Q} \in D^{|Q|}$ which in turn can be of numerical or qualitative nature specified by the domain $D$. If we query a subset of arms $Q$ for $t$ many times, then $\mathbf{o}_Q(t)$ is the corresponding feedback vector. We suppose that we are given a suitable statistic $s$ for the type of observation vectors, which maps a multiset of observations to some specific value relevant for the decision making. With this, $\mathbf{s}_Q(t) = (s_{i|Q}(t))_{i \in Q}$ is the statistic vector derived by the sequence of feedback $(\mathbf{o}_Q(t))_t$ of the query set $Q \in \mathcal{Q}_{\leq k}$, and $s_{i|Q}(t) = s(\{o_{i|Q}(1), \ldots, o_{i|Q}(t)\})$ is the relevant statistic for decision making about arm $i$ in the "context" $Q$ after querying $Q$ for $t$ many times.

**Examples.** For combinatorial bandits with semi-bandit feedback, the observation $\mathbf{o}_Q(t) = (o_{i|Q}(t))_{i \in Q}$ corresponds to the reward one obtains for each arm $i \in Q$ by using $Q$ for the $t$-th time, so that in particular $D = \mathbb{R}$. The most natural statistic in this case is the empirical mean given for a multiset $O$ of observations by $s(O) = \frac{1}{|O|} \sum_{x \in O} x$, such that $s_{i|Q}(t) = \frac{1}{t} \sum_{t'=1}^{t} o_{i|Q}(t')$, which is also the arguably most prevalent statistic used in the realm of bandit problems for guiding the decision making process. However, other statistics $s$ such as quantiles or the expected shortfall are of interest as well [13].

In the preference-based bandit setting with winner feedback we observe after the $t$-th usage of the query set $Q$ only a binary (winner) information, i.e., $o_{i|Q}(t) = 1$ if arm $i$ is preferred over the other arms in $Q$ at "pull" $t$ and $o_{i|Q}(t) = 0$ otherwise, so that $D = \{0, 1\}$. Once again the empirical mean of these binary observations is a quite intuitive choice for the statistic $s$, as in a stochastic feedback setting the corresponding statistic vector $s_{i|Q}(t) = \frac{1}{t} \sum_{t'=1}^{t} o_{i|Q}(t)$ would converge to the probability vector determining how likely an arm will be preferred over all the other arms in the query set $Q$. For preference-based bandits with full ranking feedback we observe after the $t$-th usage of the query set $Q$ an entire ranking of the arms in $Q$, i.e., $o_{i|Q}(t)$ is arm $i$'s rank among the arms in $Q$ at "pull" $t$, so that $D = \{1, \ldots, k\}$. In such a case the statistic $s$ might be a positional scoring rule [28].

**Goal.** The goal of the learner in our setting is to find a or the best arm (specified below) within a fixed budget of at most $B$ samples (numbers of queries). For any $Q$, write $n_Q(t)$ for the number of times $Q$ has been queried until (including) time $t$. An algorithm, which tackles the problem, chooses at time $t$ a set $Q_t \in \mathcal{Q}_{\leq k}$ and observes as feedback $\mathbf{o}_{Q_t}(n_{Q_t}(t))$ leading to an update of the relevant statistic vector $\mathbf{s}_{Q_t}(n_{Q_t}(t)) = (s_{i|Q_t}(n_{Q_t}(t)))_{i \in Q_t}$.

**Best arm.** Inspired by the theoretical groundings of Hyperband [25, 34] for best arm identification problems in numerical bandit problems with non-stochastic rewards, we make the following assumption regarding the limit behavior of the statistics

$$(A1) : \forall Q \in \mathcal{Q}_{\leq k} \; \forall i \in Q \; : \; S_{i|Q} := \lim_{t \to \infty} s_{i|Q}(t) \text{ exists.}$$

This assumption is in general slightly looser than assuming (stationary) stochastic feedback mechanisms, as (A1) is fulfilled for many prevalent statistics by means of a limits theorem such as the strong law of large numbers. Conceptionally, our Assumption (A1) is similar to the assumption on the sequence of losses in [25], as both have in common that the statistics (losses in [25]) converge to some fixed point, respectively. However, due to the difference of the action spaces (single arms vs. set of arms) and the nature of the feedback (scalar vs. vector observation), our assumption can be seen as a combinatorial extension of the one in [25].

Given assumption (A1) a straightforward notion of a best arm is obtained by leveraging the idea of a Borda winner from dueling bandits with pairs of arms as the possible subsets to more general subsets of arms. A *generalized Borda winner* (GBW) is then an arm which has on average the largest asympotical statistic, i.e.,

$$i_{\mathcal{B}}^* \in \arg\max_{i \in \mathcal{A}} S_i^{\mathcal{B}} = \arg\max_{i \in \mathcal{A}} \frac{\sum_{Q \in \mathcal{Q}_{=k}(i)} S_{i|Q}}{|\mathcal{Q}_{=k}(i)|},$$

where $\mathcal{Q}_{=k}(i) = \{Q \in \mathcal{Q}_{=k} \mid i \in Q\}$ and $S_i^{\mathcal{B}}$ are the asymptotic Borda scores, i.e., the limits according to (A1) of $s_i^{\mathcal{B}}(t) := \frac{\sum_{Q \in \mathcal{Q}_{=k}(i)} s_{i|Q}(t)}{|\mathcal{Q}_{=k}(i)|}$. Similarly, a *generalized Copeland winner* (GCopeW) is an arm $i$, which wins w.r.t. the asymptotic statistics on average on the most query sets, i.e.,

$$i_{\mathcal{C}}^* \in \arg\max_{i \in \mathcal{A}} S_i^{\mathcal{C}} = \arg\max_{i \in \mathcal{A}} \frac{\sum_{Q \in \mathcal{Q}_{=k}(i)} \mathbf{1}\{S_{i|Q} = S_{(1)|Q}\}}{|\mathcal{Q}_{=k}(i)|}.$$

However, both these notions of best arm have two major drawbacks, as there might be multiple GBWs and GCopeWs and due to averaging over all subsets in their definition, there is no way to identify a GBW or a GCopeW within a sampling budget of $o\left(\binom{n-1}{k-1}\right)$ in the worst case (see Theorem 3.1).

In light of these drawbacks, we specify another reasonable notion of a best arm, for which we leverage the concept of the *generalized Condorcet winner* [21, 4] from the preference-based bandits literature. For this purpose, we introduce the following assumption

$$(A2): \exists i^* \in \mathcal{A} \text{ such that } \forall Q \in \mathcal{Q}_{\leq k}(i^*) \ \forall j \in Q \setminus \{i^*\} \text{ it holds that } S_{i^*|Q} > S_{j|Q},$$

where $\mathcal{Q}_{\leq k}(i) = \{Q \in \mathcal{Q}_{\leq k} \mid i \in Q\}$ for $i \in [n]$. We call $i^*$ the generalized Condorcet winner (GCW), which is the arm dominating all the other arms in each possible query set containing it. It is worth noting that such an arm may not exist, but if it exists, then it is arguably the most natural way to define the optimal arm, even though it may differ from the GBW. Nevertheless, the existence of the generalized Condorcet winner (or simply the Condorcet winner for the case $k = 2$) is a common assumption in the preference-based bandits literature [4, 21, 8]. Additionally, we will show below that identifying a GCW is possible for a sampling budget of size $\Omega(n/k)$ even in worst case scenarios.

**Problem characteristics.** In light of (A1) and (A2), there are two key characteristics which will determine the appeal of any learner in our setting. The first one is the speed of convergence of the statistics $s_{i|Q}$ to their limit values $S_{i|Q}$. More precisely, the function $\gamma_{i|Q} : \mathbb{N} \to \mathbb{R}$, which is the point-wise smallest non-increasing function fulfilling $|s_{i|Q}(t) - S_{i|Q}| \leq \gamma_{i|Q}(t)$ for any $t \in \mathbb{N}$, plays a major role in characterizing the difficulty of the learning problem. Moreover, the worst speed of convergence function of a query set $Q \in \mathcal{Q}_{\leq k}$ given by $\overline{\gamma}_Q(t) := \max_{i \in Q} \gamma_{i|Q}(t)$ and the overall worst speed of convergence function $\overline{\gamma}(t) := \max_{Q \in \mathcal{Q}_{\leq k}} \overline{\gamma}_Q(t)$ will be of relevance as well. Assuming a stochastic setting, the role of $\gamma_{i|Q}$ is played by the minimax rate of convergence of the statistic to its population counterpart, e.g., $1/\sqrt{t}$ for the empirical mean and the expected value. Usually the speed of convergence functions will appear implicitly by means of their (quasi-) inverses given by $\gamma_{i|Q}^{-1}(\alpha) := \min\{t \in \mathbb{N} \mid \gamma_{i|Q}(t) \leq \alpha\}$, $\overline{\gamma}_Q^{-1}(t) := \min_{i \in Q} \gamma_{i|Q}^{-1}(t)$ and $\overline{\gamma}^{-1}(t) := \min_{Q \in \mathcal{Q}_{\leq k}} \overline{\gamma}_Q^{-1}(t)$.

The other relevant problem characteristic are the gaps of the limits statistics, i.e., $\Delta_{i|Q} := S_{i^*|Q} - S_{i|Q}$ for $i \in [n]$, $Q \in \mathcal{Q}_{\leq k}(i) \cap \mathcal{Q}_{\leq k}(i^*)$. Such gaps are prevalent in the realm of bandit problems, as they can be used to define a measure of (sub-)optimality of an arm in the stochastic feedback case. Note that in our setting this is not straightforward, as the gaps are depending on the query set and more importantly the speed of convergence has a decisive impact on the severeness of these gaps.

## 3 Lower Bounds

Let us abbreviate $\mathbf{S} := (S_{i|Q})_{Q \in \mathcal{Q}_{\leq k}, i \in Q}$ and $\boldsymbol{\gamma} := (\gamma_{i|Q}(t))_{Q \in \mathcal{Q}_{\leq k}, i \in Q, t \in \mathbb{N}}$. Given $\mathbf{S}$ and $\boldsymbol{\gamma}$, write $\mathfrak{S}(\mathbf{S}, \boldsymbol{\gamma})$ for the set of all $\mathbf{s} = (s_{i|Q}(t))_{Q \in \mathcal{Q}_{\leq k}, i \in Q, t \in \mathbb{N}}$ that fulfill

(i) $\forall Q \in \mathcal{Q}_{\leq k}, i \in Q$: $S'_{i|Q} = \lim_{t \to \infty} s_{i|Q}(t)$ exists,
(ii) $\forall Q \in \mathcal{Q}_{\leq k}, i \in Q, t \in \mathbb{N}$: $|s_{i|Q}(t) - S'_{i|Q}| \leq \gamma_{i|Q}(t)$,
(iii) $\forall Q \in \mathcal{Q}_{\leq k}$: $\exists \pi_Q : Q \to Q$ bijective such that $S'_{i|Q} = S_{\pi(i)|Q}$ for all $i \in Q$.

For $Q$ and $l \in \{1, \ldots, |Q|\}$ write $S_{(l)|Q}$ for the $l$-th order statistic of $\{S_{i|Q}\}_{i \in Q}$, i.e., $\{S_{i|Q}\}_{i \in Q} = \{S_{(l)|Q}\}_{l \in Q}$ and $S_{(1)|Q} \geq \cdots \geq S_{(|Q|)|Q}$. If Alg is a (possibly probabilistic) sequential algorithm, we denote by $B(\text{Alg}, \mathbf{s})$ the number of queries made by Alg before termination when started on $\mathbf{s}$. In the following, we provide lower bounds on $\mathbb{E}[B(\text{Alg}, \mathbf{s})]$ for algorithms Alg, which identify, for any instance $\mathbf{s} \in \mathfrak{S}(\mathbf{S}, \boldsymbol{\gamma})$, almost surely one GCW resp. GBW resp. GCopeW of $\mathbf{s}$.

**Theorem 3.1.** *Let $\mathbf{S}$ be such that $S_{(1)|Q} > S_{(2)|Q}$ for all $Q \in \mathcal{Q}_{\leq k}$.*
*(i) There exists $\mathbf{s} \in \mathfrak{S}(\mathbf{S}, \boldsymbol{\gamma})$ such that if Alg correctly identifies a GCW for any instance in $\mathfrak{S}(\mathbf{S}, \boldsymbol{\gamma})$, then*

$$\mathbb{E}\left[B(\text{Alg}, \mathbf{s})\right] \geq \left\lceil \frac{n}{k} \right\rceil \min_{Q \in \mathcal{Q}_{\leq k}, j \in Q} \gamma_{j|Q}^{-1}\left(\frac{S_{(1)|Q} - S_{(|Q|)|Q}}{2}\right).$$

*(ii) Assume $(S_{(1)|Q}, \ldots, S_{(|Q|)|Q})$ does not depend on $Q$ for any $Q \in \mathcal{Q}_{=k}$. If Alg correctly identifies a GBW for any instance in $\mathfrak{S}(\mathbf{S}, \boldsymbol{\gamma})$, then*

$$\sup_{\mathbf{s} \in \mathfrak{S}(\mathbf{S}, \boldsymbol{\gamma})} \mathbb{E}\left[B(\mathrm{Alg}, \mathbf{s})\right] = \Omega\left(\binom{n-1}{k-1}\right).$$

*(iii) If Alg correctly identifies a GCopeW for any instance in $\mathfrak{S}(\mathbf{S}, \boldsymbol{\gamma})$, then it fulfills*

$$\sup_{\mathbf{s} \in \mathfrak{S}(\mathbf{S}, \boldsymbol{\gamma})} \mathbb{E}\left[B(\mathrm{Alg}, \mathbf{s})\right] = \Omega\left(\binom{n-1}{k-1}\right).$$

The theorem is proven in Section B in the supplement, where we provide in fact slightly stronger versions of these bounds.

## 4 Algorithms

In this section, we present a class of algorithms along with three possible instantiations solving the corresponding learning task for the case of the generalized Condorcet winner being the best arm. In particular, we analyze all algorithms theoretically in terms of their sufficient and necessary budget to find the respective best arm. In light of the results in Theorem 3.1 for GBW and GCopeW identification, it is straightforward that a simple algorithm, which enumerates all possible subsets, pulls each of them equally often in a round-robin fashion, and returns the empirical GBW (or GCopeW) is already optimal. For sake of completeness, we show this result for the case of GBW identification in Section C and also include this simple algorithm in the experimental study below (called ROUNDROBIN).

### 4.1 Generalized Condorcet Winner Identification

In the following we introduce a general class of algorithms in Algorithm 1 which is instantiable with various different elimination strategies of the arms. Below, we present some instantiations which build on commonly used elimination strategies in the standard multi-armed bandit setting. The idea of Algorithm 1 is simply to maintain a set of active arms, which is successively reduced by following a specific arm elimination strategy (Algorithm 2) referred to as elimination rounds. In each elimination round $r \in \{1, 2, \ldots, R\}$ the set of active arms $\mathbb{A}_r$ is partitioned into $P_r$ many sets of size $k$ (up to a possible remainder set) denoted by $(\mathbb{A}_{r,j})_{j \in P_r}$ for which the elimination strategy is applied with a roundwise-dependent budget $b_r$. The budget allocated to a partition $\mathbb{A}_{r,j}$ in round $r$ is of the form $b_r = \lceil B/(R \cdot P_r) \rceil$ following the idea to split up the available budget equally first for each round and second for all partitions in each round. The explicit arm elimination strategy used in Algorithm 1 is specified by Algorithm 2 and corresponds to pulling the chosen query set $Q$ for a fixed number of times and afterwards keeping only the best $f(|Q|)$ arms of $Q$. Here, $f : [k] \to [k]$ is an arbitrary function with $f(x) \leq x - 1$ for all $x$, which essentially determines the aggressiveness or conservativeness of an arm elimination strategy as we will see below.

---

**Algorithm 1** Combinatorial Successive Elimination

**Input:** set of arms $[n]$, subset size $k \leq n$, sampling budget $B$, a function $f : [k] \to [k]$, sequence $\{P_r\}_r$ (number of partitions at round $r$), $R$ (number of rounds in total)
**Initialization:** $\mathbb{A}_1 \leftarrow [n], r \leftarrow 1$

1: **while** $|\mathbb{A}_r| \geq k$ **do**
2:     $b_r \leftarrow \lfloor B/(P_r R) \rfloor$, $J \leftarrow P_r$
3:     $\mathbb{A}_{r,1}, \mathbb{A}_{r,2}, \ldots, \mathbb{A}_{r,J} \leftarrow \mathrm{Partition}(\mathbb{A}_r, k)$
4:     **if** $|\mathbb{A}_{r,J}| < k$ **then**
5:        $\mathcal{R} \leftarrow \mathbb{A}_{r,J}, J \leftarrow J - 1$
6:     **else**
7:        $\mathcal{R} \leftarrow \emptyset$
8:     **end if**
9:     $\mathbb{A}_{r+1} \leftarrow \mathcal{R}$
10:    **for** $j \in [J]$ **do**
11:       $\mathcal{R} \leftarrow \mathrm{ArmElimination}(\mathbb{A}_{r,j}, b_r, f(|\mathbb{A}_{r,j}|))$
12:       $\mathbb{A}_{r+1} \leftarrow \mathbb{A}_{r+1} \cup \mathcal{R}$
13:    **end for**
14:    $r \leftarrow r + 1$
15: **end while**
16: $\mathbb{A}_{r+1} \leftarrow \emptyset$
17: **while** $|\mathbb{A}_r| > 1$ **do**
18:    $\mathbb{A}_{r+1} \leftarrow \mathrm{ArmElimination}(\mathbb{A}_{r+1}, b_r, f(|\mathbb{A}_{r+1}|)), r \leftarrow r + 1$
19: **end while**

**Output:** The remaining item in $\mathbb{A}_r$

---

In the following we provide three possible instantiations of Algorithm 1 inspired by commonly used elimination strategies in the standard multi-armed bandit setting for pure exploration tasks.

**Combinatorial Successive Winner Stays.**
The most aggressive elimination strategy is to keep only the arm with the best statistic (the winner) of each partition in each round and discard all others from the set of active arms for the next round. Concretely, we use $f^{\mathrm{CSWS}}(s) = 1$ for $f$ in Algorithm 1 in this case.

| **Algorithm 2** ArmElimination$(\mathbb{A}', b, l)$ |
| --- |
| 1: Use $\mathbb{A}'$ for $b$ times |
| 2: For all $i \in \mathbb{A}'$, update $s_{i\mid\mathbb{A}'}(b)$ |
| 3: Choose an ordering $i_1, \ldots, i_{\mid\mathbb{A}'\mid}$ of $(s_{i\mid\mathbb{A}'}(b))_{i\in\mathbb{A}'}$ |
| 4: **return** $\{i_1, \ldots, i_l\}$ |

The resulting instantiation of Algorithm 1 is called *Combinatorial Successive Winner Stays* (CSWS), which has at most $R^{CSWS} = \lceil \log_k(n) \rceil + 1$ many rounds in total (at most $\lceil \log_k(n) \rceil$ rounds in the first while-loop and at most 1 in the second). The total number of partitions in round $r$ is at most $P_r^{CSWS} = \lceil n/k^r \rceil$.

**Combinatorial Successive Reject.** On the other extreme regarding the aggressiveness of the arm elimination strategy is to dismiss only the worst arm of each partition in each round and keep all others in the set of active arms for the next round. More specifically, we use $f^{\mathrm{CSR}}(s) = s - 1$ for this variant, which can be seen as a variant of the Successive Reject algorithm [5] for best arm identification adopted to the combinatorial bandit problem. Consequently, we call the resulting instantiation of Algorithm 1 the *Combinatorial Successive Reject* (CSR) algorithm, whose number of rounds in the first while-loop is at most $\lceil \log_{k^{-1}/k} (1/n) \rceil$ and in the second at most $k - 1$. Overall, we have a maximal number of rounds $R^{CSR} = \lceil \log_{k^{-1}/k} (1/n) \rceil + k - 1$ and a maximal number of partitions per round $P_r^{CSR} = \lceil n(1-\frac{1}{k})^{(r-1)}/k \rceil$.

**Combinatorial Successive Halving.** As a compromise between the aggressive elimination strategy of CSWS and the conservative elimination strategy of CSR one could discard in every elimination round the worse half of all arms in the partition, i.e., using $f^{\mathrm{CSH}}(s) = \lceil s/2 \rceil$ for $f$ in Algorithm 1. This can be seen as a generalization of the successive halving algorithm [25] adopted to the combinatorial bandit problem we are considering. Thus, the instantiation of Algorithm 1 in this spirit will be called the *Combinatorial Successive Halving* (CSH) algorithm. Note that we have at most $\lceil \log_2(n) \rceil$ rounds in the first while-loop and additional $\lceil \log_2(k) \rceil$ in the second while-loop resulting in at most $R^{CSH} = \lceil \log_2(n) \rceil + \lceil \log_2(k) \rceil$ many rounds throughout a run of CSH. Furthermore, we have at most $P_r^{CSH} = \lceil \frac{n}{2^{r-1}k} \rceil$ partitions in round $r$.

## 4.2 Theoretical Guarantees

In the following, we derive the sufficient budget for Algorithm 1 to return under assumptions (A1) and (A2) the best arm $i^*$, i.e., the generalized Condorcet winner. For this purpose, we write $\mathbb{A}_r(i^*)$ for the unique set $\mathbb{A}_{r,j} \in \{\mathbb{A}_{r,1}, \ldots, \mathbb{A}_{r,P_r}\}$ with $i^* \in \mathbb{A}_{r,j}$ and define $\Delta_{(l)\mid Q} = S_{i^*\mid Q} - S_{(l)\mid Q}$ for any $Q \subseteq [n]$ with $i^* \in Q$.

**Theorem 4.1.** *Assume $P_r$, $R$ are such that Algorithm 1 called with $B$ does not exceed $B$ as total budget. Under Assumptions (A1) and (A2) Algorithm 1 returns $i^*$ if $B$ is larger than*

$$z\left(f, R, \{P_r\}_{1\le r\le R}\right) := R \max_{r\in[R]} P_r \cdot \left\lceil \bar{\gamma}^{-1}_{\mathbb{A}_r(i^*)} \left( \Delta_{(f(\mid\mathbb{A}_r(i^*)\mid)+1)\mid\mathbb{A}_r(i^*)}/2 \right) \right\rceil$$

The following theorem indicates optimality of $z$ in the theorem above (cf. Sec. D.2 for the proofs).

**Theorem 4.2.** *For any distinct asymptotic values $\mathbf{S}$, there exists a family of statistics $\{s_{i\mid Q}(t)\}_{t\in\mathbb{N}, Q\in\mathcal{Q}_{\le k}, i\in Q}$ with $s_{i\mid Q}(t) \to S_{i\mid Q}$ for all $i \in [n], Q \in \mathcal{Q}_{\le k}$ such that if Algorithm 1 is used with a budget $B < z\left(f, R, \{P_r\}_{1\le r\le R}\right)$ then it does **not** return $i^*$.*

By means of Theorem 4.1, we can infer the following result regarding the sufficient sampling budget $B$ for the three instantiations to output $i^*$ (cf. Sec. D.3 of the appendix for the proof).

**Corollary 4.3.** *Under Assumptions (A1) and (A2), CSX $\in \{$CSWS, CSR, CSH$\}$ returns $i^*$ if it is executed with a budget $B \ge z_{\mathrm{CSX}}$, where $z_{\mathrm{CSX}} := z\left(f^{\mathrm{CSX}}, R^{\mathrm{CSX}}, \{P_r^{\mathrm{CSX}}\}_{1\le r\le R^{\mathrm{CSX}}}\right).$*

By substituting the concrete values for $P_r$, $R$ and $f$ of the corresponding instantiation into Corollary 4.3 and using a rough estimate for the inverse function of the speed of convergence, we see that all of

the resulting sufficient budgets are essentially $\tilde{\mathcal{O}}(n/k)$ (see Table 1) almost[1] matching the dependency on $n$ and $k$ in Theorem 3.1. If we would allow the special case of singleton sets of arms as query sets, i.e., $k = 1$, the sufficient budget for CSH matches the one derived in [25] for its non-combinatorial counterpart in the special case of numerical feedback.

Table 1: Sufficient budget for CSWS, CSR and CSH. Here, $\pi$ is as in (iii) in Section 3.

| | |
|---|---|
| $z_{CSWS}$ | $\left\lceil \frac{n}{k} \right\rceil (\lceil \log_k(n) \rceil + 1) \cdot \max_{Q \in \mathcal{Q}_{\leq k} : i^* \in Q} \max_{i \in Q \setminus \{i^*\}} \left\lceil \bar{\gamma}^{-1} \left( \frac{S_{i^*|Q} - S_{i|Q}}{2} \right) \right\rceil$ |
| $z_{CSR}$ | $\left\lceil \frac{n}{k} \right\rceil \left( \left\lceil \log_{1-\frac{1}{k}} \left( \frac{1}{n} \right) \right\rceil + k - 1 \right) \cdot \max_{Q \in \mathcal{Q}_{\leq k} : i^* \in Q} \min_{i \in Q \setminus \{i^*\}} \left\lceil \bar{\gamma}^{-1} \left( \frac{S_{i^*|Q} - S_{i|Q}}{2} \right) \right\rceil$ |
| $z_{CSH}$ | $\left\lceil \frac{n}{k} \right\rceil (\lceil \log_2(n) \rceil + \lceil \log_2(k) \rceil) \cdot \max_{Q \in \mathcal{Q}_{\leq k} : i^* \in Q} \left\lceil \bar{\gamma}^{-1} \left( \frac{S_{i^*|Q} - S_{\pi(Q)|Q}}{2} \right) \right\rceil$ |

Regarding $n$ and $k$ both lower and upper bounds coincide, but the gap-term in the lower bounds include a min-term over $\mathcal{Q}_{\leq k}$, while the gap-term in the upper bound are coming with a max-term over $\mathcal{Q}_{\leq k}$. The difference between these terms depends on the underlying hardness of the bandit problem in terms of $\bar{\gamma}^{-1}$, i.e., how fast the considered statistics converge to their limit values. Due to the generality of our setting it is difficult to specify this difference more explicitly and it would be worth considering this for special cases, i.e., the numerical bandits or preference-based bandits separately.

Finally, it is worth mentioning that all of the three instantiations of Algorithm 1 have only been studied for the case of single arm pulls, but not for pulls of subsets of arms, where additionally a dependency on the set might be present. Thus, the theoretical guarantees are novel in this regard.

## 5   Applications to Stochastic Settings

**Numerical feedback.** In stochastic combinatorial bandits [16], each arm-query set pair $(i, Q)$ is associated with a probability distribution $\nu_{i|Q}$ and querying $Q$ for the $t$-th time results in the feedback $o_{i|Q}(t) \sim \nu_{i|Q}$, usually referred to as a reward (i.e., $D = \mathbb{R}$). The sequence of rewards $\{o_{i|Q}(t)\}_t$ is supposed to be independent and the statistic $s$ is the empirical mean such that (A1) holds by the law of large numbers with $S_{i|Q} = \mathbb{E}_{X \sim \nu_{i|Q}}[X]$. If the $\nu_{i|Q}$ are sub-Gaussian, an anytime confidence bound by [24] based on the law of iterated logarithm ensures $|s_{i|Q}(t) - S_{i|Q}| \leq c_\delta(t)$ for all $t \in \mathbb{N}$ with probability at least $1 - \delta$ for some appropriate function $c_\delta(t) \in \mathcal{O}(\sqrt{t \ln(\ln(t)/\delta)})$. This implies the following result, the proof of which is deferred to Section E.

**Corollary 5.1.** *Let $f, R$ and $\{P_r\}_{r \in [R]}$ be as in Theorem 4.1 and suppose the reward distributions $\nu_{i|Q}$ to be $\sigma$-sub-Gaussian and such that their means $S_{i|Q}$ satisfy (A2). There is a function $C(\delta, \varepsilon, k, R, \sigma)$ in $\mathcal{O}\left(\sigma^2 \varepsilon^{-2} \ln\left(kR/\delta \ln\left(kR\sigma/\varepsilon\delta\right)\right)\right)$ such that if $i^*$ is the optimal arm for $(S_{i|Q})_{Q \in \mathcal{Q}_{\leq k}, i \in Q}$ and $\sup_{Q \in \mathcal{Q}_{\leq k}(i^*)} \Delta_{(f(|Q|)+1)|Q} \leq \varepsilon$, then Algorithm 1 used with a budget $B$ larger than $C(\delta, \varepsilon, k, R, \sigma) \cdot R \max_{r \in [R]} P_r$ returns $i^*$ with probability at least $1 - \delta$.*

**Other statistics for numerical feedback.** A rich class of statistics can be obtained by applying a linear functional $U(F) = \int r(x)\mathrm{d}F(x)$, where $F$ is a cumulative distribution function (CDF) and $r : \mathbb{R} \to \mathbb{R}$ some measurable function [49], on the empirical CDF, i.e., $\tilde{s}(O, x) = |O|^{-1} \sum_{o \in O} \mathbf{1}\{x \leq o\}$, for any $x \in \mathbb{R}$ and any multiset of (reward) observations $O$. This leads to the statistics

$$s_{i|Q}(t) = U(\tilde{s}(o_{i|Q}(1), \ldots, o_{i|Q}(t), \cdot)) = \sum_{s=1}^{t} \frac{r(o_{i|Q}(s))}{t},$$

which converge to $S_{i|Q} = \mathbb{E}_{X \sim \nu_{i|Q}}[r(X)]$ by the law of large numbers, provided these expected values exist. For this class of statistics we can show a quite similar result to Corollary 5.1 by generalizing the findings in [24] (see Section E). However, the result is in fact more general than Corollary 5.1 as for $r$ being the identity function we can recover the empirical mean.

---

[1] Here, $\tilde{\mathcal{O}}$ hides logarithmic factors.

**Preference feedback.** In the preference-based bandits, we observe when querying $Q$ for the $t$-th time a categorical random variable with values in $Q$, i.e., $o_{i|Q}(t) \sim \text{Cat}_Q(\mathbf{p}_Q)$ for some underlying unknown parameter $\mathbf{p}_Q = (p_{i|Q})_{i \in Q}$. Let $w_{i|Q}(t) := \sum_{s \leq t} \mathbf{1}\{o_{i|Q}(t) = i\}$ be the number of times arm $i$ has won in the query set $Q$ until the $t$-th pull of $Q$. We consider as the relevant statistics $s_{i|Q}(t) = \frac{w_{i|Q}(t)}{t}$, which converge to $p_{i|Q} =: S_{i|Q}$ by the law of large numbers. The Dvoretzky-Kiefer-Wolfowitz inequality [19] ensures a concentration inequality on $\sup_{i \in Q} |s_{i|Q}(t) - S_{i|Q}|$, which can be used to deduce the following result (cf. Sec. E for the proof).

**Corollary 5.2.** *Let $f, R$ and $\{P_r\}_{r \in [R]}$ be as in Theorem 4.1 and suppose preference-based winner feedback with parameter $(p_{i|Q})_{Q \in \mathcal{Q}_{\leq k}, i \in Q}$, which satisfies (A2). There is a function $C(\delta, \varepsilon, k, R) \in \mathcal{O}\left(\varepsilon^{-2} \ln\left(R/\delta\varepsilon^4\right)\right)$ with the following property: If $i^*$ is the optimal arm and $\sup_{Q \in \mathcal{Q}_{\leq k}(i^*)} \Delta_{(f(|Q|)+1)|Q} \leq \varepsilon$, then Algorithm 1 used with a budget $B$ larger than $C(\delta, \varepsilon, k, R) \cdot R \max_{r \in [R]} P_r$ returns $i^*$ with probability at least $1 - \delta$.*

By substituting the concrete values for $P_r$, $R$ and $f$ of the corresponding instantiation of Algorithm 1 into the bound on the budget in Corollary 5.2 (compare to Table 1), we see that each of the three resulting bounds almost matches the optimal sample complexity bounds for identifying the (generalized) Condorcet Winner under fixed confidence in preference-based bandits [8, 21] indicating near optimality of the algorithms in stochastic settings. However, since no stochastic counterpart of our combinatorial setting for the numerical case exists, it would be interesting to investigate whether the analogous implication by means of Corollary 5.1 for the three algorithms is nearly optimal as well. We leave this to future work, as it is beyond the scope of our work.

## 6 Experimental Section

In this section we present an experimental study for our proposed algorithms on an algorithm selection problem. Further experiments, also on synthetic data and with other statistics are provided in the supplementary material in Section G.

**Setting.** In the following, we consider an algorithm selection problem, where the goal is to select the most efficient algorithm for solving an instance of a satisfiability (SAT) problem. For this, we randomly chose $n = 20$ parameterizations of the SAPS solver [23] which represent our candidate algorithms and correspond to the arms in our terminology. Our possible problem instances are sampled from the first 5000 problem instances from the sat_SWGCP folder of the AClib[2]. We compare CSWS, CSR, CSH and ROUNDROBIN on this problem with the Successive Halving (SH) algorithm [25]. To the best of our knowledge, there are no algorithms available as baselines, which are designed for the pure exploration problem with finite budget and subsets of arms as the actions, e.g., [4] investigates a regret minimization problem, while [21] is dealing with a stochastic pure exploration setting with fixed-confidence. However, Successive Halving serves as a baseline, which we included as a representative for the algorithms dealing with a pure exploration problem with finite budget and single arms as the actions. In each learning round, we randomly draw a problem instance from the 5000 problem instances without replacement and then start a parallel solution process with the SAPS parameterizations chosen by the corresponding learning algorithm (only one parameterization for the case of SH), where the process is stopped as soon as the first algorithm has solved the current instance. In particular, one obtains only for the "finisher" SAPS parameterization an explicit numerical value (its runtime) among the chosen set of SAPS parameterizations, as the others are right-censored. Since our proposed algorithms are designed for the case, in which feedback for all arms in the pulled query set is observed, while SH is designed for the case in which only a single arm is queried resulting in a single feedback, we enlarge the available budget for SH to $k \cdot B$ for a fairer comparison.

**Instantiation of CSE.** Although we could consider the negative runtimes of the parameterizations as rewards (i.e., runtimes correspond to losses) and use a statistic suitable for numerical feedback for the combinatorial successive elimination (CSE) approaches, there might be a major disadvantage due to the censored runtimes. Indeed, in order to apply a statistic suitable for numerical feedback, some sensible imputation technique is required to deal with the censored observations, which in turn could introduce a severe bias. However, thanks to the generality of our framework, we can simply

---

[2]http://www.aclib.net

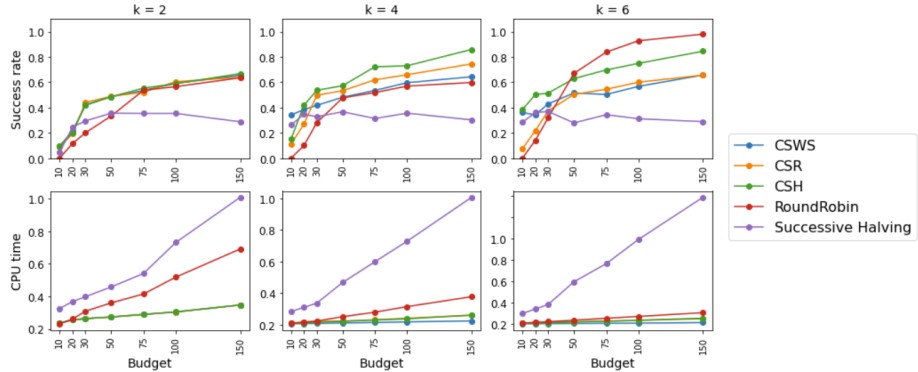

Figure 1: Success rates and runtimes for different subset sizes $k$ and budgets $B$.

interpret the observed feedback as a preference in the sense that the "finisher" SAPS parameterization is preferred over the others in the chosen set of parameterizations. In this way, using a statistic based on preference-based feedback defuses the bias issue. Quite naturally, we use the relative frequency statistic for preference feedback as specified in Section 5.

**Analysis.** As the best arm (SAPS parameterization) we use the one having the smallest empirical runtime over all problem instances such that ROUNDROBIN and SH will tendentially return this arm if the budget is sufficiently large. The resulting success rates for our proposed algorithms and SH of identifying the best arm are illustrated in the top panel of Figure 1. One can see, that the algorithms which follow our CSE strategy significantly outperform SH if the budget is sufficiently large. In addition, CSWS, CSR and CSH identify the best arm more often than ROUNDROBIN if the subset sizes $k$ are small, which is a realistic situation in practice. Moreover, the bottom panel in Figure 1 shows the overall runtimes of the algorithms revealing that SH takes much longer than CSWS, CSR and CSH and as expected the difference in the runtimes gets larger with the subset sizes $k$. Quite interestingly, even ROUNDROBIN needs a longer runtime than the CSE approaches, although it queries the same number of subsets and also stops the respective run as soon as the "finisher" SAPS parameterization is clear. Thus, the differences in the runtimes of ROUNDROBIN and the CSE approaches are only due to the fact that the latter discard the slowest SAPS parameterizations quickly and do not run them again, while ROUNDROBIN uses throughout all subsets the same amount of time, even if they contain only bad performing parameterizations. In other words, the differences are due to the sophisticated strategies of the CSE approaches.

# 7   Future Work

For future work, it would be interesting to investigate whether switching the elimination strategy during the learning process leads to any performance improvements both theoretically and empirically. A similar question could be asked regarding the considered statistic for numerical feedback variants. Further, the goal of identifying the best set of arms in our scenario would also be interesting. However, in the case where the observations depend on the chosen set of arms, it is far from obvious how to define a suitable optimality term (cf. Sec. 6.3.2 in [8]). Finally, a more extensive experimental study would definitely be a relevant future direction of research, especially for hyperparameter optimization problems with possible parallelization options such as in [33] or for more general algorithm configuration problems [44].

## Acknowledgments and Disclosure of Funding

This work was partially supported by the German Research Foundation (DFG) within the project "Online Preference Learning with Bandit Algorithms" (project no. 317046553) and by the research training group "Dataninja" (Trustworthy AI for Seamless Problem Solving: Next Generation Intelligence Joins Robust Data Analysis) funded by the German federal state of North Rhine-Westphalia.

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
