# OpenReview forum: "Finding Optimal Arms in Non-stochastic Combinatorial Bandits with Semi-bandit Feedback and Finite Budget"
_NeurIPS.cc/2022/Conference — NeurIPS 2022 Accept_

### Official Review · Reviewer_pjkv · 2022-07-08

**Rating:** 6
**Confidence:** 3
**Soundness:** 3 good
**Presentation:** 3 good
**Contribution:** 3 good

**Summary:**

This paper considers the combinatorial bandits problem with semi-bandit feedback and finite budget constraint, and introduces novelty by considering non-stochastic and subset-dependent feedback. Towards solving the problem the paper introduces algorithms based on successive elimination and provides corresponding lower bounds on the problem. Numerical experiments demonstrate the power of these algorithms in different contexts.

**Questions:**

N/A. See weakness section for more details.

**Limitations:**

See weaknesses for more details.

**Strengths And Weaknesses:**

Strengths:
1. The paper is overall well-organized, with each section clearly stating what it does. Theoretical guarantees on upper bounds are also provided for each algorithm in discussion, and intuition is overall clear with corresponding lower bounds.
2. The experiments are indicative of the theoretical claims made in the paper. Sufficient details have been provided to make reproduction possible.
3. Related work and future work are also included with comprehensive discussion.

Weaknesses:
1.Some theorems could be better formatted in the paper. For instance, the section on lower bounds is a bit hard to read with the lack of paper spacing.
2. It would be helpful if the authors could further explain the relationship between the lower bounds and the respective upper bounds the algorithms are able to achieve, as the statements in their current forms are not immediately clear.

---

> ### Author Response · Authors · 2022-08-02
> **Answer to Reviewer pjkv**
>
> Thank you for your helpful and valuable comments. We are very pleased that you appreciate the impact of our theoretical results and that you find our paper well-organized. We will address your open questions and concerns in the following.
>
> ### Bad formatting of theorems.
>
> Yes, we admit that this is a bit short due to space constraints. If our paper gets accepted, we will use the additional page to reformat the theorems.
>
> ### Relationship between lower and upper bounds.
>
> We will extend our discussion by the following: Regarding $n$ and $k$ both lower and upper bounds coincide, but the lower bound includes a min-term over the gaps, while the upper bound results in Theorem 4.1 are coming with a max-term over the gaps. The difference between these constants depends on the underlying hardness of the bandit problem in terms of $\bar{ \gamma}^{-1}$, i.e., how fast the considered statistics converge to their limit values. Due to the generality of our setting it is difficult to specify this difference more explicitly and it would be worth considering this for special cases, i.e., the numerical bandits or preference-based bandits separately. It is worth noting that the upper bounds we achieve for a stochastic setting are nearly matching the lower bounds (see Section 5) for both variants.

---

> > ### Comment · Reviewer_pjkv · 2022-08-07
> > **Response**
> >
> > Thank you for your comments. I will keep my score.

---

### Official Review · Reviewer_8B47 · 2022-07-10

**Rating:** 6
**Confidence:** 2
**Soundness:** 3 good
**Presentation:** 2 fair
**Contribution:** 2 fair

**Summary:**

This paper studies a bandit problem where at each time $t\in\left\lbrace 1,...,B \right\rbrace$, the decision maker can query a subset of size  at least $2$ and at most $k$ from a collection of $n$ arms (naturally, $k \leqslant n$). Upon query, a semi-bandit feedback is realized which comprises of an observable for each arm in the query set. This observable could be a real-valued reward, a 0/1-valued outcome indicating the ``winner,'' or the rank w.r.t. other arms in the query set. The semi-bandit feedback could be generated by an oblivious adversary.

The authors make two assumptions on the underlying dynamics. (A1) Fixing an arm and a query set, some test function (statistic) of the sequence of observations from said arm converges to a well-defined asymptotic limit. This is a fairly general condition that settings with, e.g., i.i.d. or ergodic rewards naturally satisfy. The second assumption (A2) posits the existence of a generalized Condorcet Winner (GCW), which is the arm dominating all the other arms in each possible query set containing it. Such an arm need not necessarily exist but the authors claim there are antecedents in the literature that rely on this assumption.

Subject to this premise, the authors study the Best Arm (GCW) Identification problem in the fixed budget setting with $B$ samples. Lower and upper bounds are investigated for this problem setting (the latter for separate instantiations of a successive elimination-based meta-algorithm). Results are evaluated in the special case of stochastic combinatorial bandits; a more general setting is evaluated numerically.

**Questions:**

The paper is generally well-written, but it would really help readers who are not entirely abreast of developments in this area if the actual optimization problem that the authors set out to solve is stated formally and clearly at some point. This would also make it easier to interpret the lower and upper bounds derived in the paper.

W.r.t. the algorithmic proposition, each of the three routines that the meta-algorithm can be instantiated with is well-studied in the literature. Other than that, am I missing any innovation? I would like to see the authors discuss the benefits of their framework (if any) other than providing a unified higher-level abstraction.

**Limitations:**

The paper is written in a way that this aspect appears to be generally well-addressed.

**Strengths And Weaknesses:**

To my knowledge, the BAI problem for combinatorial semi-bandits under (oblivious) adversarial feedback is certainly novel. However, I am not very well-positioned to evaluate the technical innovations in this work vis-\`a-via the extant body of work in the area. Though I do have a couple of broad comments (see the next section).

---

> ### Author Response · Authors · 2022-08-02
> **Answer to Reviewer 8B47**
>
> Thank you for your helpful and valuable comments. We will address your open questions and concerns in the following.
>
> ### Actual optimization problem that is solved.
>
> Please note that our goal is stated in the paragraph **Goal** in Section 2 and this is exactly the optimization problem we seek to solve. Moreover, please note that this definition is in line with the common definition in the bandit literature with fixed budget, see [1] or [2].
>
> ### Innovations and benefits of our framework.
>
> Please note that the three instantiations of our framework are only studied for the case of single arm pulls, but not for pulls of subsets of arms, where additionally a dependency on the set might be present. Thus, the theoretical guarantees are novel in this regard.
> In addition, the theoretical analysis is rather non-standard due to the combinatorial setup of the problem. In particular, the lower bounds results are novel and the proofs are non-trivial.
> Last but not least, it is worth mentioning that the unification allows to consider instead of numerical feedback also preference-based feedback, which has recently been successfully applied in other domains [3,4] and is also the key to obtain satisfactory results in our experimental study (this is mentioned in lines 73-79 in the introduction).
>
> $[1]$ Bubeck et al. 2009. Pure Exploration for Multi-Armed Bandit Problems.
> $[2]$ Audibert et al. 2010. Best arm identification in multi-armed bandits.
> $[3]$ Kirschner et al. 2021. Bias-robust bayesian optimization via dueling bandits.
> $[4]$ Mohr et al. 2021. Single player Monte-Carlo tree search based on the Plackett-Luce Model.

---

### Official Review · Reviewer_q5ks · 2022-07-21

**Rating:** 7
**Confidence:** 3
**Soundness:** 4 excellent
**Presentation:** 3 good
**Contribution:** 3 good

**Summary:**

The authors consider a generalization of the best arm identification problem under a fixed budget setting. The agent seeks to identify the best base arm, but can take actions of subsets (up to cardinality $k$) each time and receive semi-bandit feedback.  The feedback’s properties (like base arm expected values for stochastic setting) can vary with the subset chosen, though some ordering (in the sense of a Condorcet winner) is assumed across all possible subsets.  Additionally, the authors generalize the domain of the feedback, encompassing both (numerical) reward feedback and preference feedback (like dueling bandits).  With this, the authors identify lower bounds on the sampling budget, propose several successive elimination type algorithms (varying in eliminating just the worst, all but the best, or the worst half) and identify their corresponding upper bounds on the sufficient budget.  The authors also run experiments on both numerical reward feedback and preference feedback problems.

**Questions:**

- lines 281-283 mention if $k=1$, sufficient budget for CSH matches that for [24], though is that is only if singletons are used (ie CSH can only pick individual arms as opposed to allowing it pick subsets of arms of cardinality $\ell=1,\dots,k$)?  For preference feedback it may be vacuous to consider allowing single arm actions (when $k>1$), but for numerical feedback there may be cases where it could be beneficial.  Though perhaps not helpful enough to affect bounds without additional assumptions.

Minor clarifications
- While the basic assumption that statistics have limits (A1) is general (allowing statistics for the same arm $i$ to vary considerable based on query set $Q$, under (A2) that there is a (generalized) Condorcet winner more or less reduce to the ‘Choice model’ proposed in [4]?


**Limitations:**

Yes

**Strengths And Weaknesses:**


Strengths (major)
- This paper addresses a novel setting – (oblivious) non-stochastic BAI with combinatorial actions under a fixed budget, and is general enough to cover both preference and numerical (reward) vector feedback.  [I think the contribution is especially significant for settings with preference feedback]
- Lower bounds on the budget are provided and upper bounds analyzed for three algorithms (all partition and remove arms, but vary in how aggressively remove arms)
- Overall the paper is well-written
- The supplementary material covers a number of additional helpful points
- I did not carefully review proofs; that being said I did not spot any issues with the results


Strengths (minor)
- The allowance for statistics to vary based on the query set appears more general than (many) prior works that assuming latent parameterized preference models like Plackett-Luce or even randomized utility models.  (though assumption (A2) restricts the statistics to have important structure for making the problem tractable that is present in earlier models)

Weaknesses (major)

-[I did not identify any major weaknesses]

Weaknesses (minor)
- The setting considered appears well motivated for preference feedback, but I think needs more motivation for numerical feedback.  In particular, (1) the problem of looking for only a single base arm when actions can consist of subsets of arms, (2) the definitions of ‘best arms’ as a Borda, Copeland, or Condorcet winner (3) the feedback vector assumption, and (4) the agent cannot pick a single arm $Q={i}$, the minimum cardinality is 2, all appear to be natural properties for problems that motivate BAI with preference vector feedback.  For numerical feedback, however, works on pure exploration with combinatorial action spaces consider identifying the best action (ie subset), since subsets could have larger rewards than base arms.
- The baseline used in experiments, RoundRobin, does not exploit (A2)-- the experiments would have been stronger if other/additional baselines had been used.  In particular, using algorithms assuming stochastic feedback with preference vector feedback and that also exploit (A2), perhaps those of [4] or [21].  They may suffer against oblivious adversaries, but could provide much more informative baselines for experiments.

---

> ### Author Response · Authors · 2022-08-02
> **Answer to Reviewer q5ks**
>
> Thank you for your helpful and valuable comments. We are very pleased that you appreciate the generality of our framework and that you find our paper well-written. We will address your open questions and concerns in the following.
>
> ### Lacking motivation for numerical feedback.
>
> Please note that such a setting is also considered in existing works, please see also lines 62-79. However, without taking dependencies into account. Moreover, the practically motivated AC problem we consider is also of such a kind.
>
> ### Other baselines than RoundRobin.
>
> To the best of our knowledge, there are no algorithms available as baselines, which are designed for the pure exploration problem with finite budget and subsets of arms as the actions, e.g., [1] investigates a regret minimization problem, while [2] is dealing with a stochastic pure exploration setting with fixed-confidence.
>
> However, please note that Successive Halving, which we included in our experiments, is another baseline, which we included as a representative for the algorithms dealing with a pure exploration problem with finite budget and single arms as the actions.
>
> ### For $k=1$ the sufficient budget for CSH matches that for [3].
>
> Yes, this is correct. Also your statement that this would be vacuous for the preference-based bandit case and this is why we exclude the case $k=1$ in our setting.
>
> ### Does the assumption of a generalized Condorcet Winner reduce the setting to that of choice bandits?
>
> Only if we would additionally assume a stochastic environment, the feedback is preference-winner feedback, and if the statistic $s$ is the relative winning frequency.
>
> $[1]$ Agarwal et al. 2020. Choice bandits.
> $[2]$ Haddenhorst et al. 2021. Identification of the generalized Condorcet winner in multi-dueling bandits.
> $[3]$ Jamieson et al. 2016. Non-stochastic best arm identification and hyperparameter optimization.

---

### Meta-Review · Area_Chair_HWa7 · 2022-08-24

**Recommendation:** Accept
**Confidence:** Certain

**Metareview:**

The reviewers came to consensus that this paper has a good contribution to the study of pure exploration for the combinatorial bandits. On the other hand, several minor concerns such as the motivation, experiments and algorithmic novelty, are raised. I agree that these concerns are reasonable and please polish the manuscript by addressing these points in the final version.

**Award:**

No

---

### Decision · Program_Chairs · 2022-09-14

Accept